# Blurred Edges: Representation of Space in Transgenerational Memory of the Nazi Euthanasia Program

Erika Silvestri [1,2] 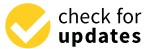

1 Zentrum für Antisemitismusforschung, TU Berlin, 10553 Berlin, Germany; erika.silvestri@uniroma1.it
2 Dipartimento SARAS, Università La Sapienza, 00185 Roma, Italy

**Abstract:** Maria Fenski was born on 14 August 1905, in Papenburg. At the age of seventeen, she was diagnosed with "dementia" and hospitalized at the Provinzial-Heil-und Pflegeanstalt Osnabrück, where she remained until 16 January 1923. After a marriage, three children, some happy family years, and various commitments to different clinics, she was killed in Neuruppin State Institution in Brandenburg in 1942, as one of the people murdered in the Nazi Euthanasia Program. Her granddaughter, Hannah, produced a series of sixteen paintings dedicated to her grandmother's story. There are almost no people in Hannah's artwork, but empty, lonely, symbolic spaces able to create a bond between past and present. The lack of human figures, the use of cold colors and the blurred edges contribute to creating a suspended atmosphere that seems to be full of painful silences and negations. Hannah transferred onto the canvas an echo of the feelings the victims could have felt, living in conditions they could not understand, separated from the world before they were each made to face a solitary death, far from any contact with their families. Analyzing her work, I reflect on the importance of the concept of "Space" in this specific transgenerational transmission of "*Aktion T4*" family memory.

**Keywords:** *Aktion T4*; Nazi euthanasia; transgenerational memory; transgenerational trauma; collective memory

## 1. Introduction

Maria Fenski (Figure 1) is a victim of the Nazi Euthanasia Program. Born in 1905, she was diagnosed with "dementia praecox" in 1922 and admitted to the Provinzial-Heil-und Pflegeanstalt Osnabrück, where she remained until January 1923. After a marriage and some years of happiness, including the birth of three children, and after numerous hospitalizations over the years, in 1939, the family's physician ordered her to be interned at the Evangelisches Krankenhaus Königin Elisabeth Herzberge clinic in Berlin, Lichtenberg with the diagnosis of "schizophrenia". From this clinic, on 10 June 1940, a request was made for her sterilization, but was refused by the competent Gesundheit Gericht following her husband's protest. In 1941, Maria was transferred to Neuruppin, where she died a year later, on 7 August 1942. Her stated cause of death was a "heart attack," although she had never had any previous heart problems.

National Socialist Germany was not the first regime to introduce sterilization by law,[1] thus making it a compulsory measure for people suffering from serious hereditary diseases. By the end of the 1920s, 28 US states had already introduced laws providing for sterilization for criminals, alcoholics, and individuals considered "genetically dangerous", e.g., intellectually disabled people, people with epilepsy, or, in some cases, with physical deformities (Kühl 2013). The preparatory phase of the Nazi Euthanasia Program was the creation of a legislative apparatus that made sterilization compulsory for certain categories of people and prohibited procreation between "Aryans" and individuals of "different races" (Weindling 1989, p. 441 and following). On 14 July 1933[2], the Minister of the Interior, Wilhelm Frick, issued the *Gesetz zur Verhütung erbkranken Nachwuchses*, a law on the

prevention of the birth of persons suffering from hereditary diseases, which came into force in January 1934, and provided for the forced sterilization of individuals with a hereditary physical or mental illness (Bock 1986). Each case was to be examined by a Hereditary Health Court consisting of two physicians and a district judge, who, if successful, would order the sterilization without the need for the patient's consent. Hereditary Health Appeals Courts were also set up, with the same composition and a final decision, to which anyone found positive for sterilization could appeal. Götz Aly writes of 350,000 individuals sterilized in the first seven years of the National Socialist Reich (Aly 2017, p. 4). The order initiating the "merciful death" operations was only put in writing in the autumn of 1939,[3] although it had previously been issued informally. Hitler backdated it to 1 September: the start of the Second World War is directly linked to the killing of individuals with disabilities, which was a precondition for the extermination of the Jews and other persecuted minorities (Kühl 2013, pp. 124–25). The authorization signed by Hitler served as a sort of legal basis, because no law on euthanasia was ever enacted by Nazi Germany. On 18 August 1938, the *Reichsministerium des Innern* issued a strictly confidential decree,[4] ordering all doctors and midwives to register by means of a declaration all children under the age of three in whom serious hereditary diseases were suspected.[5] The Reich Committee had the task[6] of collecting the data and submitting them to three experts, W. Catel, E. Wentzler, and H. Heinze. If all three agreed to proceed, the patient was admitted to one of the specialist departments set up for the killing. In the summer of 1939, Hitler also entrusted K. Brandt and P. Bouhler with the responsibility of killing adults. The project's management offices were located at Tiergartenstrasse 4 in Berlin, which is why the adult euthanasia program was given the code name "*Aktion T4*". Six institutes were set up for this purpose: Brandenburg an der Havel, Grafeneck, Hartheim, Pirna-Sonnenstein, Bernburg an der Saale, and Hadamar. After the arrival, patients were made to undress, then measured and weighed. A doctor carried out a very brief examination of each one to decide on a probable cause of death based on the state of health. Unlike the children, no one asked for the consent of adult patients or permission from their families, neither for the transfer nor, of course, for the killing (Aly 2013). Families received one letter from the institution when the transfer had already taken place and two from the killing center: the notice of arrival (when the patient had in fact already been killed), with a request not to visit their relative, and the notice that death and cremation had already taken place. If relatives requested it, the institutions sent them the urn with random ashes. All death certificates were signed by doctors using pseudonyms: the crime had to remain secret. In the first phase of the program, from January 1940 to August 1941, a total of more than 70,000 people were killed in those suppression clinics (Friedlander 1995, p. 25). On 23rd August 1941, Hitler issued the order to suspend adult euthanasia operations with immediate effect from the following day, but the killing of children and adults continued with the creation of special wards. More than 10,000 children and young people were killed between 1939 and 1945 (Aly 2017, p. 75). The euthanasia plan entered its so called "wild phase", in which the doctors took total control of the decision-making process, even to the point of being able to determine the methods of killing.

　　In this essay, I will present an example of how the memory of this crime has been preserved in the families of the victims through the history of Maria. Firstly, using excerpts from interviews with two of Maria's granddaughters, Hannah and Gina, I will delineate how the memory of the crime has been inherited by the next generations and which role they played in the transmission. To do this, I draw extensively from quotes from the interviews I have personally recorded. One could easily object and argue that they are too extensive, but I consider it important to give space to the direct voice of the victim's granddaughters. It is indeed their voice that can clearly show, without the filter of my interpretation, the ways in which Maria's memory has been preserved in her family, the difficulties and omissions with which it has been transmitted, and the traumas it has generated.

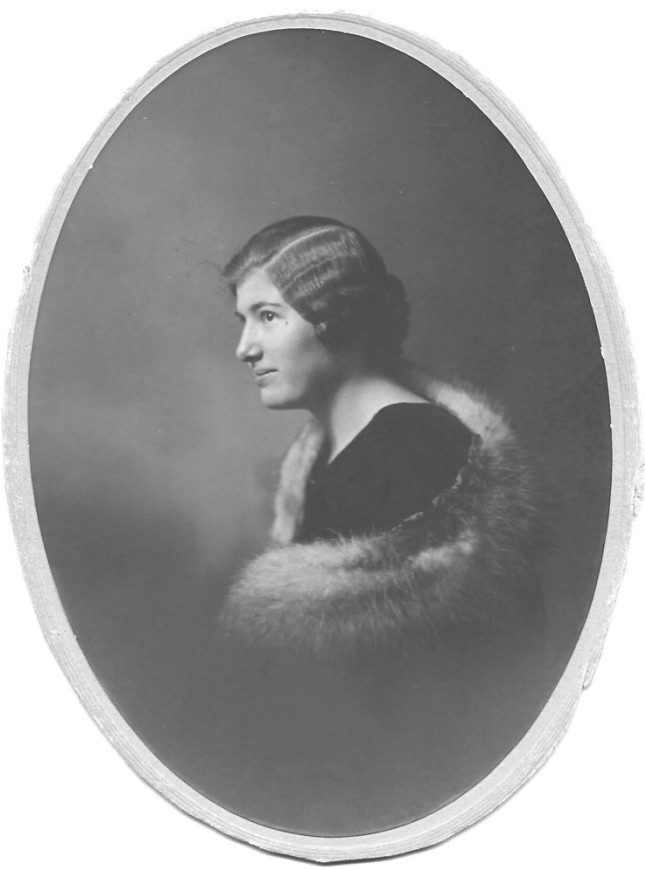

**Figure 1.** Maria Fenski, 1927.

Hannah Bischof, whom I met and interviewed in Berlin, produced a series of sixteen paintings dedicated to her grandmother's story. I will analyze her artwork in the second part of this essay, focusing on the central role she gave to the use of space and its meaning. There are almost no people in Hannah's work, but empty, lonely, symbolic spaces able to create a bond between past and present. The lack of human figures, the use of cold colors, and the blurred edges contribute to creating a suspended atmosphere that seems to be full of painful silences and negations. Hannah transferred onto the canvas an echo of the feelings the victims could have felt, living in conditions they could not understand, separated from the world before they were each made to face a solitary death, far from any contact with their families. As true "memory objects", the paintings have high symbolic value, manifesting at the same time the existence of the victim, her history, and the decisive role of the granddaughter, who brought the memory back to life. To understand their symbolic value, it is important to emphasize that they are not merely containers of a private memory but constitute themselves as individualities, as subjects. It is precisely as subjects that they act, enabling remembrance, thus mediating between oblivion and presence and connecting a victim's family memory to the collective memory of Nazi euthanasia.

## 2. Transgenerational Memory in the Family of a Nazi Euthanasia Victim

Maria's history, like that of so many other victims of the Nazi Euthanasia Program, is a story of pain and misunderstanding, loneliness and silence. By condensing it into a few sentences full of dates, diagnoses, and hospitalizations, there is a risk of forgetting that she was a human being. Reading and reflecting on how she was degraded, to the point of being killed, is completely useless if we are not always able to perceive, behind the words, the woman she was, first as a daughter and then as a mother, who loved and was loved by her family; a woman who had the right to have a full and peaceful life.

The risk of losing the human being behind the enormity of the crime committed often also affects the members of the victims' own families. In fact, annihilation was not merely a physical matter but involved the reduction of the individual to an elemental stage, in which the name (as was the case in the extermination camp system) was replaced by a number and one's image completely denied.[7] When the relatives of a victim of the Nazi Euthanasia Program attempt to reconstruct his or her story, they find it extremely difficult to imagine the victim's own perception of reality during the periods of internment and at the moment of death. Not having been able to share their memories in the first person, victims have in fact been denied the opportunity to provide their own testimony. In the case presented here, this voice is provided by Maria's granddaughter: "I have the feeling of perceiving what my grandmother experienced. I imagine and feel what was going on inside her."[8]

Maria was born on 14 August 1905, in Papenburg. On 17 October 1922, at the age of seventeen, she was diagnosed with "dementia praecox" and admitted to the Provinzial-Heil-und Pflegeanstalt Osnabrück, where she remained until 16 January 1923. At the end of June 1927, she married Josef Fenski. On 9 September 1928, after the birth of her first son, she was hospitalized again with the diagnoses of "schizophrenia" and "wochenbettpsychose" (post-partum depression) at the Staatskrankenanstalt Friedrichsberg Hamburg, where she remained until 26 February 1929. She led a happy life for about ten years, and she moved with her husband to Berlin, where two other children were born in 1930 and 1935. On 21 August 1938, she was again admitted, with the diagnosis of "paranoid psychosis", to the private sanatorium Heidehaus Zepernick, near Bernau, from which she was discharged on 24 September. Following numerous crises, she was admitted and discharged from the same institution several times, until the family's physician ordered her to be admitted, with the diagnosis of "schizophrenia", to the Evangelisches Krankenhaus Königin Elisabeth Herzberge clinic in Berlin, Lichtenberg, where she arrived on 24 June 1939. On 10 June 1940, a request was made for her sterilization, but was refused by the competent Gesundheit Gericht on 6 September 1940, after the protest of her husband. On 14 August 1941, she was transferred to Neuruppin, where she died a year later, on 7 August 1942, with the stated cause of death being a "heart attack".

I met Maria's granddaughters, Hannah and Gina, for the first time in Berlin,[9] on a rainy day in June 2019. They had both worked on the reconstruction of their grandmother's story, so I initially interviewed them together in order to better understand the role they both played. Then, I met Hannah several times, the painter, to record individual interviews. I have used an open interview style, leaving space for free narrative, with the only indication to follow (at least at the beginning) a chronological narrative structure.[10] In the first phase, I asked a few questions necessary to generate the narrative. In subsequent meetings, after working on the victim's story, I asked more specific questions to clarify key points.

Hannah, born in 1960, graduated in law and worked for several years as a lawyer before deciding to devote herself to painting. Gina, born in 1962, studied biology; she also specialized in genetics and now works as an educator. I interviewed both sisters in Hannah's atelier, a small room in the back of a ceramicist's gallery, filled with paintings leaning against the wall, a wooden easel, various objects, and a small table to sit at. Above a shelf full of cups and glasses, resting perhaps in wait for guests or visitors, a large black and white photograph of Maria hangs on the wall. It is impossible not to notice it. I returned several times to read all the documentation these women had collected over the years, which allowed me to date each stage of Maria's story very precisely. They gathered medical records and other hospital files, private letters, court documents. The strong bond between the two sisters, which they defined as a "special relationship", was immediately apparent. Both confirmed that they have three brothers with whom they do not share the same affinity. Their answers to my questions about Maria's story and about their family followed each other harmoniously, very often one completing the other's sentences. To learn more about the family context and the affective relationships existing between the generations, I started asking about their grandparents. Maria was the mother of their father, but both sisters did not know much about her when they were young. Hannah and Gina's father

was only a child when Maria died, and the woman they knew as their grandmother was their grandfather's second wife. Gina linked Maria's illness to her family's dysfunctional situation, interpreting it as somehow connected; Hannah, on the other hand, managed to fit what she discovered into the history of the Third Reich.

> Gina:[11] *I was about 18 years old, at the time. I didn't know there was a secret in the family, until my father said that his mother was mentally ill, suffering from schizophrenia. I laughed, thinking, "What a family! I have an eccentric grandmother, and you never told me about her!", I didn't understand the true significance at the time. I was in a fight with my father and didn't consider how he must have felt, his feelings. This realization came later. I couldn't imagine how he felt as a child, so I couldn't recognize what was behind the story. My father was seven years old when she died, but he had not seen her before either, as she was often hospitalized and, in 1940, was admitted in a special closed section and never came home.*

> Hannah: *I found out earlier, when I was 16 years old. He told me she had died and I asked if she was murdered by the Nazis, as I had heard about that in school. We talked about euthanasia. He answered, "Nobody knows", as she had a disease. When we found her medical file, it says (showing the document) that she died of a "heart attack". But last year, searching on the internet, I learned that this was often written when a patient died of starvation.*

> G: *But she was really schizophrenic.*

Although he was motherless at the age of seven, the father of my interviewees never stopped wanting to understand how and why his mother had died. After he stopped working, during his retirement years, he started his research, writing to the clinics where his mother had been hospitalized. However, he did not receive any useful information.

> G: *Our father knew that she died in Neuruppin. He had this document about her, the "Totenschein" (death certificate), but he was not sure about the cause of death. He had two siblings and one older sister who was five years older. She asked their father, after the war, if it could be that she died of heart disease, as she had never had one before, so she thought their mother was killed. But our father, who was five years younger, wasn't so sure. So, in the family, they didn't think she was murdered, but that she was . . . brought to death, how to say. They had no real evidence. Our father started searching for her file, I think it was around 1970, and wrote to all the hospitals where she had been. He was the first one who wanted to know the truth.*

> H: *It was after the Berlin Wall came down that he wrote to the clinic in Berlin Lichtenberg, where she was from 1938 to 1941, the Evangelisches Krankenhaus Königin Elisabeth Herzberge, but they said, "We have nothing. No files about her". Then, he wrote to Neuruppin, which also had no information.*

When Gina decided to continue the research started by her father, she received different answers from the clinics. In 2003, she wrote again to Neuruppin, and this time they invited both sisters and gave them a copy of the files.

> H: *The Director of the clinic spoke with us for two or three hours and said we were the first people to ask for files. It was in 2003.*

> G: *I knew my father had ask before, and I am sure that many others had done the same! The Director was born after the war, so it wasn't his fault. But he felt that he was representing the hospital and he knew the Doctor who wrote the death certificate, he was still alive! I don't know if he really was the murderer of my grandmother, I can't say for sure. But he signed the documents and it was his responsibility to take care of the patients, to ensure that they were living in the proper way . . .*

> H: *. . . and not be killed.*

> G: *Yes, to take care of them. So, I started to do research, and asked Hannah to join me. I didn't want to go to Neuruppin alone. I felt very . . . it was so . . . cold. To visit that*

*place knowing that my grandmother died there . . . under such circumstances, starving and left alone by the doctors and others who were supposed to take care of her . . . but she was not the only one. Many people died there. It was very difficult to visit that place and see all those windows closed, like a prison.*

I asked the two sisters to tell me what they had discovered and reconstructed.

H: *When Maria left Papenburg and her family, she went to Hamburg.*

G: *She gave birth to her first child and suffered from postpartum depression. For half a year, she stayed in a clinic, and our grandfather went to Berlin to work for the Christian workers' syndicate. He was very religious and made a career for himself. Berlin was the centre of that organization and he earned some money. He decided to take his wife back with him and rented a house in the suburbs, with a little garden. There, their second child arrived. It was a nice life . . .*

H: *For almost ten years, life was nice. Then she was hospitalized again in a clinic.*

G: *. . . but times changed. She started to go back and forth between the clinic and home. It was a private hospital in Berlin, near Bernau. I think it was just too much for her.*

H: *She had the feeling of always being observed by someone behind her, looking for her.*

G: *It's all written in her personal file.*

H: *Then, in 1938, the family's doctor declared that she was schizophrenic and had to be hospitalized again. The doctor was not an expert, just a regular one. It was the summer of 1939.*

H: *The doctor at the clinic requested her sterilization, but she was not sterilized. Our grandfather went to the Gesundheit Gericht to claim against it. It was the same place where I had my first case when I was a lawyer!*

G: *Yes, the same place! It's strange . . . it's so strange! I also worked in a laboratory in the same clinic in Lichtenberg where Maria was. It was a lab about eugenics . . . so strange.*

H: *In the end, they declared that sterilization was not necessary because she already had three children and would have no more.*

H: *I think the family's doctor knew about the euthanasia program.*

G: *He had to. There was a law, it was impossible to say "no".*

H: *The problem, in our family, was that our grandfather went to war early, in 1939, right at the beginning. He had three children and no one to take care of them.*

G: *He had to find someone to take care of them. Their mother was in the clinic. There were different women, one after another . . . it must have been very difficult for the children.*

H: *But he came back soon. Somehow, he was able to stay in Berlin. Maybe because of the children had no one to take care of them? We don't know.*

G: *He wrote a lot of letters to the doctors. We have them all in the files. It was very interesting to read them: "What's going on with my wife?", "Is she ok?", "When will she come back?". He was concerned, and visited her regularly, on Sundays, but without the children, in Neuruppin.*

G: *Then, we found another file, the first one, with all the descriptions of how she was behaving. It was very cruel . . . she was so upset and wanted to take her life. She was full of fears. I think something very bad must have happened to her.*

H: *We couldn't explain it. One day she went to Church to talk to the priest and confess. After two weeks, she said what she told him wasn't right. Did she lie? Then, her father took her to a clinic, saying she was looking for a knife to kill herself. It's all written in the file. But we don't believe her feelings of guilt were generated by the "false confession". There must have been something . . . something happened, for sure.*

G: *. . . an attack on her. She was only 17 years old . . . maybe in a sexual way? That could be the reason.*

H: *But these are just our thoughts. We have no proof.*

Was Josef aware of the risks taken by his wife? Did he know anything about *Aktion T4*? Hannah and Gina emphasized an important element: the trust in medical professionals.

G: *She was really ill. I think he couldn't understand that she could be in danger and even be killed there. He just trusted the doctors.*

H: *They believed in physicians. When a doctor said, "You must do this", then they did it. This was the strategy of the Nazi.*

G: *And it worked. Most people didn't realize that until the moment when the papers came: "Your wife died of a heart attack", but she never had heart problems! Then, people started asking . . .*

H: *This was the time when T4 program was officially stopped, but it continued secretly. Von Galen spoke about it. He said that the Nazis would kill even the wounded soldiers, and so people started to stand up.*

During the "wild phase" of the euthanasia program, when killing operations were decentralized and delegated to smaller clinics, Maria was probably slowly killed by starvation.[12] A document proves it: in the Neuruppin personal file, the weight chart lacks the final part, relating to the final year.

G: *She was losing weight while she was in the hospital. There's a weight table in the file.*

H: *She lost 30 kilos in just three years.*

G: *She lost weight until she weighed just 42 kilos. They let her die of starvation. But this page was missing from the file that Neuruppin gave us! An historian found it later and gave it to us.*

H: *I don't think the missing part was a mistake. I don't think so.*

Maria had three children when she was admitted to the clinic at Lichtenberg in 1939. They surely suffered due to the estrangement from their mother, not being able to understand what was happening to her. Maria was once visited by her eldest child, who was "horrified" by the place and the hospital atmosphere. After the family was notified of Maria's death, Josef managed, incredibly, to have his wife's body released, which was then not cremated but buried in Berlin. In the patient's personal file, the autopsy declared that cremation was "not done for technical reasons".

Maria's fears seem to have been passed down to her children and grandchildren, who have long feared being diagnosed as schizophrenic. Somehow, it is precisely the illness and the anxiety it engendered that link the sisters' lives to their grandmother's life.

G: *It has always been difficult for our family to deal with this illness. Some relatives told us, "You will get the same". So . . . you have to think about it when you get married and have children, because they said it's hereditary.*

H: *I was always afraid of getting it. When I was younger, this was the connection to Maria. I often had depression and my thoughts, especially when I was 17 or 18 years old, were: "If I tell someone I'm feeling bad, they'll take me to a clinic, and in the clinic, they'll kill me". So, I said nothing, but there was the fear.*

G: *So, we don't talk about it. Only within the family, but not to others.*

H: *When I wanted to get pregnant, I went to a doctor to ask about the risk, and my brother did too, but we didn't talk about it to each other. It was a problem in the family.*

G: *Everyone dealt with it on their own.*

H: *We were afraid. Our aunt, Maria's daughter, was always afraid, because of that. In school, when the teacher asked about her mother, she didn't mention the clinic because she was afraid of being sterilized too: "If people find out that I, a blond girl with blue eyes, a perfect Aryan type, have a mentally ill mother... with a genetic disease!"*

It is their research that has changed things: by being able to understand how and why Maria was killed and by setting her story in the historical context of Nazi Germany, Hannah and Gina were able to start to process their inherited trauma. Hannah used painting to express her grandmother's rediscovered story, which had an almost cathartic function, and Gina got to know, through her research, relatives with whom she had never had contact with before.

### 3. Zyklus für Maria as a "Memory Object"

A "Memory Object" is the concrete, sharable, and usable result of the elaboration of a transgenerational memory related to a crime by a third or fourth-generation relative of a victim. With this interpretive idea, my aim was to identify all literary, visual, figurative, musical, or theatrical expressions—or artistic expression in the broadest sense—produced by the relatives of victims; in my specific case study, this involved victims of the Nazi Euthanasia Program. These products fulfill a dual function: (1) they narrate a story that the authors have reconstructed and that is shared and exposed for the first time, moving from the private sphere to that of public memory, and (2) they testify to the active role of the author as a necessary agent for the discovery of a "family secret" and for the reconstruction of their murdered relative's story. The moment a "memory object" passes from the private to the public dimension, such as through publication, exhibition, or staging, it undergoes a transformation from object to subject. From that moment on, it acts as a subject: it is no longer a recipient, but it starts to mediate between oblivion and presence, inserting itself into the dynamics of public memorialization and claiming the space denied to the victim. It and not the author takes a stand for the victim, giving her a name, showing her face, and somehow bringing her back to life. With its existence, a "memory object" always claims a lack of responsibility for the crime. The emergence of many "memory objects", in my opinion, can determine a change of course in the process of creating a specific public memory, stimulating the emergence of a previously absent or weak public debate, claiming the failure to process certain events of the past and the failure to recognize a group of victims as such. Analyzing a "memory object" allows us to deeply investigate the history and traumatic past of a certain family, and it is precisely through this interpretive idea that I will examine Hannah's artistic production.

Between 2011 and 2015, without having a predetermined project, Hannah Bischof created sixteen paintings dedicated to the story of her grandmother, the "*Zyklus für Maria*".[13] On the website dedicated to her art, she defines the desire that drives her work as follows:

> "When I paint, I think about all and nothing. I follow my intuition and hope that it will lead me to a vision that I can understand and that shows me something—about myself or about the world in which I live. I don't think of an observer during this process. But when I have succeeded in creating the work and I am happy, I wish to be able to share this happiness. And my happiness would be complete if my painting triggered an involuntary (beautiful) memory in the viewer, as Marcel Proust described it in his novel "In Search of Lost Time". My painting leads the viewer further to something else—and thus perhaps also to a conversation with me. In this way, both sides reveal something of themselves and come closer to each other."[14]

The Zyklus paintings are characterized by the use of bright acrylic colors, with the predominance of light blue and blue tones, which are present in almost every work.[15] Only in five canvases do human figures appear, while the vast majority portray empty spaces. Like blurred, faceless puppets, almost shadows, the human figures represent Maria and her family members. Empty space and absence are used as instruments with which to narrate the victim's experiences in different moments of her life. The desolated space represents the absence of the victim's voice, her testimony, and somehow takes the role of her internal voice: we do not know how Maria experienced her internments, her transfers, the last months in the clinic where she was killed, or her terrible and solitary death.

The central role of place is also evident from the titles of Hannah's artwork: as many as fourteen have a precise reference to cities, hospitals, or clinics in which Maria lived. The subtitle itself, "*Von Papenburg nach Neuruppin*" ("From Papenburg to Neuruppin"), indicates the direction, the journey, and the space occupied by a life that began and ended between one pole of the sentence and another, and between a city and a psychiatric clinic. The cycle starts with three paintings with a very similar graphic structure: the figure of Maria appears in the lower part as the only human presence. Around her are buildings, which are symbolic of the phases of her life journey. The large red mill in the first painting, "*Die rote Mühle*" ("The red mill", Figure 2), for instance, represents the starting point of her life; her father, Anton Eissing, was at the time a chief in the sawmill of the shipyard. On the upper right are undefined dark buildings and a thin, blurred female figure stands motionless in the middle: Maria could not have been aware at the time, but she was between the start and the end of her life.

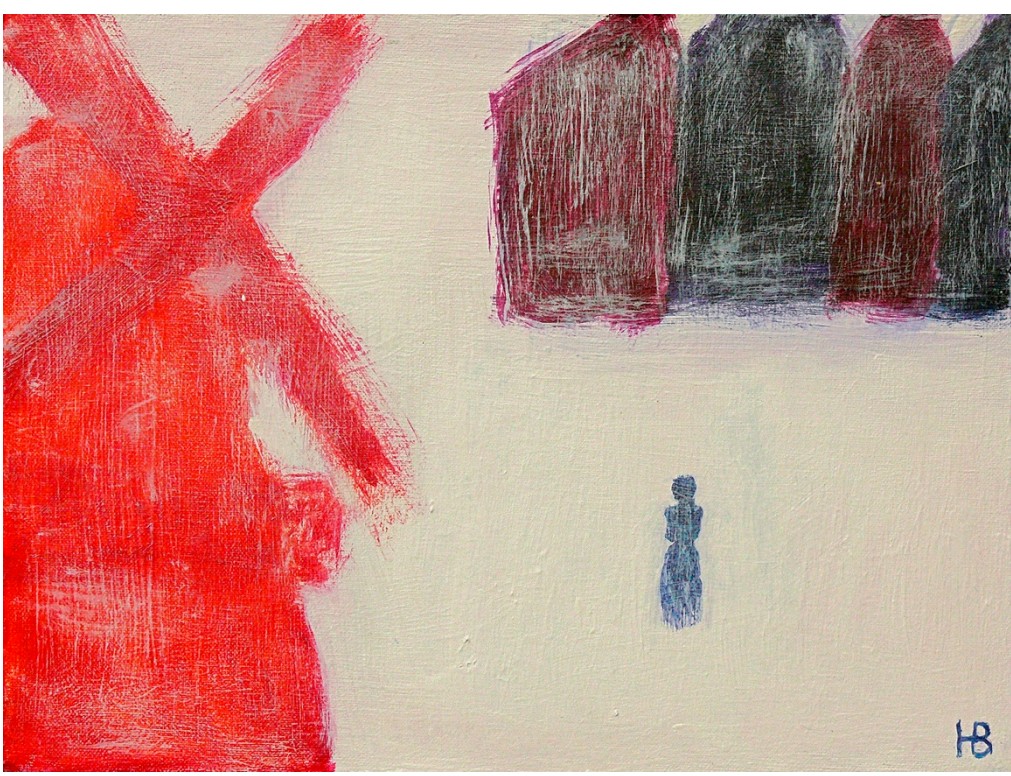

**Figure 2.** Hannah Bischof, "*Die rote Mühle*", acrylic on canvas, Berlin 2012, © Ben Bischof.

In the second work, "*Die Abreise*"[16], we see Maria at the moment of her first internment in the state hospital in Osnabrück, while in the following work, "*Papenburg 1923–1927*" (Figure 3), she is on her way back to Papenburg after being dismissed. Maria spent the years before her wedding back in Papenburg, where she worked as a "Haustochter".[17] The town was very small, people knew each other, and they used to talk about everyone: the empty, aseptic, and separating space has become larger in this work; it is almost englobing her: she is alone in front of her gloomy fate.

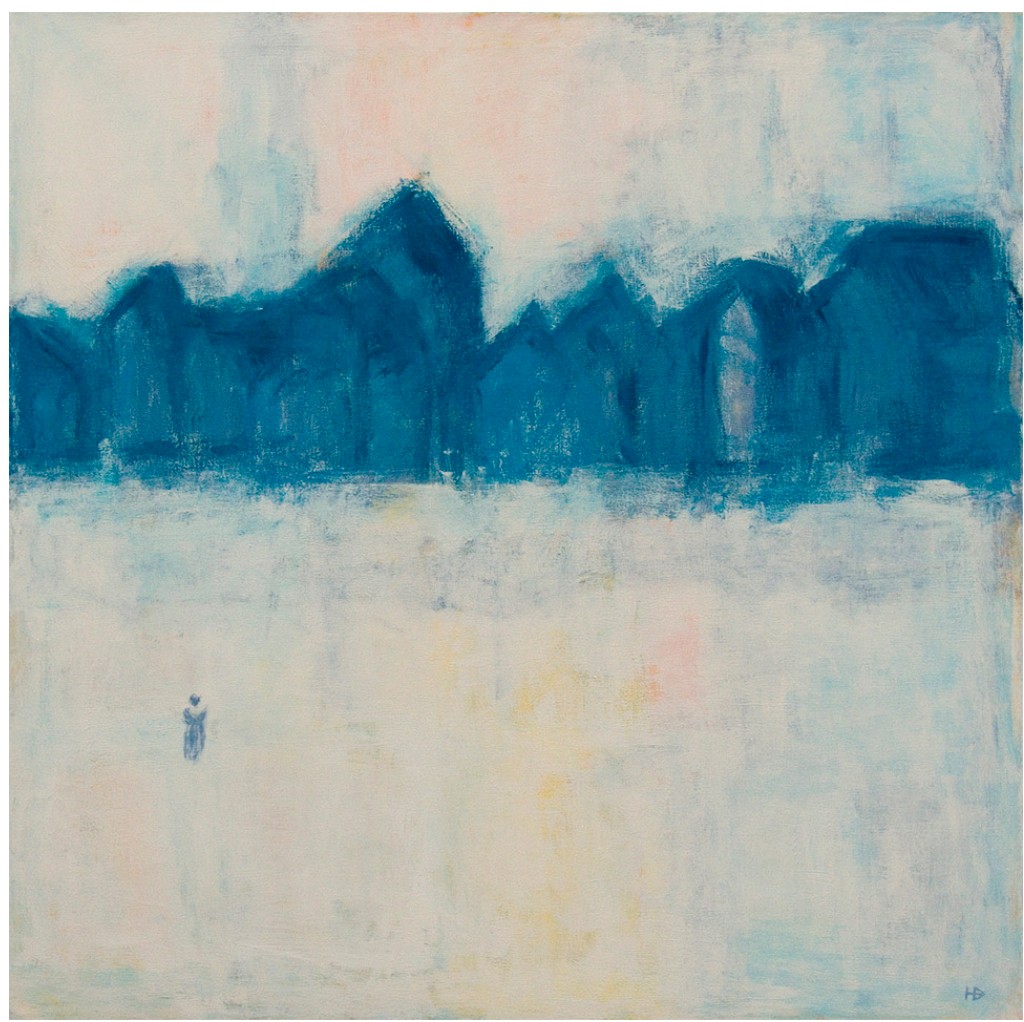

**Figure 3.** Hannah Bischof, "*Papenburg 1923–1927*", acrylic on canvas, Berlin 2015, © Ben Bischof.

Acting as a "memory object", the *Zyklus* allows us to learn about and follow Maria's story, but at the same time to perceive how her granddaughter internalized it: through Hannah's artistic elaboration, something of Maria reaches us. It is exactly as a "memory object" that the canvases become the place where the past connects with the present and allows us, for an instant, to meet the victim.

"My feeling is that Maria may have found the town cramped and even threatening after her hospitalization; she may also not have been able to talk to anyone about her stay in Osnabrück. She felt excluded, alone, abandoned. I wanted to show that fear and also that sense of being lost. This is how the idea for the painting came about: the city is dark, there is no way out, and this little person is alone in a large open space, while a dark and dangerous city awaits her on the horizon."[18]

On 25 July 1927, Maria Eissing and Josef Fenski were married. The family memory, preserved by Hannah, relates that the priest in Papenburg advised Josef against marrying Maria, because she was considered "crazy".

"When asked why Maria, Josef is reported to have replied: "Because she was the most intelligent and the most beautiful of the sisters". It is strange that the civil marriage took place in Papenburg and the church wedding in Hamburg. Maria and Joseph were strictly Catholic; the civil marriage was not considered a "proper" marriage according to the ritual of the Church; perhaps the priest in Papenburg had refused to perform the marriage precisely because Maria was considered crazy."[19]

In "*Die Hochzeit*" ("The wedding"), we can see a beautiful depiction of the event: the couple is represented as two houses next to each other, almost overlapping, and as two chairs. Although brightly tinted in yellow, the space is empty and there is no festive celebration. The element of the house is of particular importance in Hannah's work; it is used both in a metaphorical sense to represent Maria in specific circumstances of her life, and in a separative sense, i.e., as a set of walls separating inside from outside, making communication and exchange impossible. We experience the same feelings when confronted with the few photographs in circulation that can be associated with the Nazi Euthanasia Program, such as the pictures made by inhabitants of towns close to the clinics used as killing centers. Photos such as these stop at the depiction of an external and totally enclosed space, such as the clinics' buildings, which prevents us from looking beyond and grasping the reality of the crime.

"It has been remarked to me that there are many houses in my paintings and that Maria, this small human figure, is always in front of or in the middle of buildings. The house represents the feeling of being at home, the sense of protection one gets from being in one's own home. A feeling that Maria did not experience when she was in hospitals and clinics. She certainly didn't feel protected there, I think she was very afraid, and that is why in the exhibition in Osnabrück, where she was taken by her father when she was little more than a child, I decided to also display documents from her medical records, in which she is described as being very frightened and speaking in whispers, with her eyes wide open."[20]

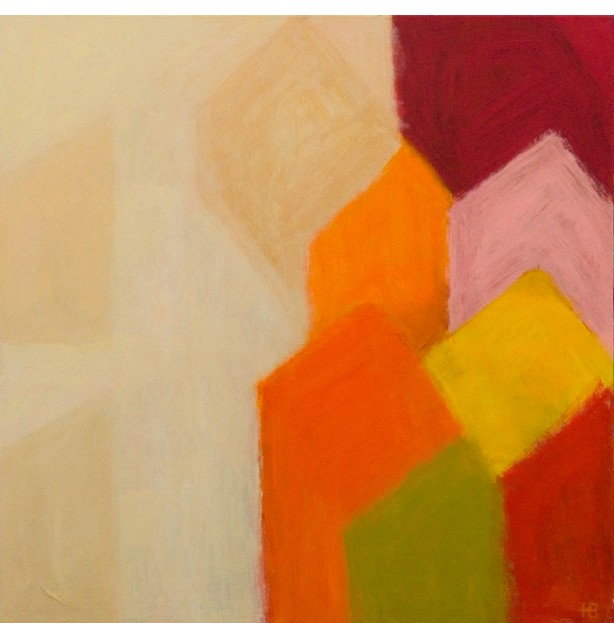

**Figure 4.** Hannah Bischof, "*Zwischenzeit*", acrylic on canvas, Berlin 2015, © Ben Bischof.

In the following works, "*Der Aufbruch*" ("The start"),[21] "*Ein neues Zuhause*" ("A new house"), "*Gestörte Wahrnehmung*" ("Disturbed perception") and "*Zwischenzeit*" ("In between", Figure 4), a positive phase in Maria's life is narrated. After their marriage, the couple left Papenburg for Hamburg, a cosmopolitan and vibrant city, where a new home was waiting for them. Hannah painted in warm colors, contrasting with the darkness of the night. A first child was born there, and, following a period of post-partum depression, Maria was interned in a clinic for about six months. In the painting, her small figure is standing at the edge of the hospital and the neighborhood where she lived; the atmosphere is suspended, the air rarefied, and the space of icy colors seems to immobilize her life. With "*Zwischenzeit*", nine years of serenity begin, without psychosis or depression. Maria was

well, the family moved to Berlin, and a daughter and another son arrived: the colors are warm, and the cold tones seem to be forgotten.

> "At that point I realized that I had to portray Maria's life, so I started to think about where she had been, which pictures to choose and I told myself that she had not always been ill, she also had an everyday life, she married, she had been loved, she had three children, she had not spent her life in hospital or in the grip of psychosis. So, I looked for photos of her, of her family, and found several. She was with Josef, or with her parents and sisters, and then of course some photos were taken in 1939, on her arrival at the Evangelisches Krankenhaus Königin Elisabeth Herzberge hospital. The classic frontal, profile, and three-quarters images were taken in those circumstances."[22]

On 21 August 1938, Maria was brought to the Heidehaus sanatorium in Zepernick, near Berlin. The diagnosis was paranoid psychosis. She was treated three times with the medication Cardiazol, which stimulates the respiratory and cardiac centers of the brain. In high doses, it causes convulsions, which is the reason why it has been used in psychiatry as shock therapy. The bright red of the house represents the effects of medicine on the heart, symbolized by the color (Figure 5); the woman almost merges with the building, so much so that she seems to disappear inside it. Her figure is the house itself, the woman's body and the institution that surrounds her have become one. From this moment on, she will never return home.

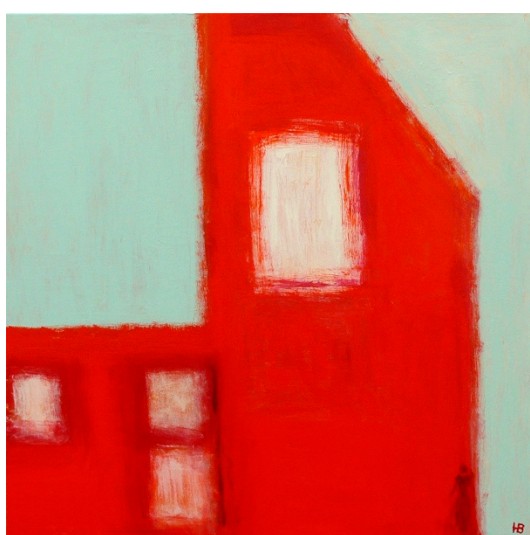

**Figure 5.** Hannah Bischof, "*Cardiazol*", acrylic on canvas, Berlin 2015, © Ben Bischof.

On 24 June 1939, Maria was admitted to the Elisabeth Herzberge hospital at the request of her family doctor. Here she remained until she was transferred to Neuruppin in 1941. There were no visitor's rooms and no café or recreation rooms; patients sat on long benches in long corridors, which opened onto windows or a glass door leading outside, to the hospital's avenues.

In the work "*Die Konturen verschwimmen—Psychiatrie*" ("The outlines blur —Psychiatry"), we can hardly recognize a bed, perhaps the inside of a hospital room. The contours are blurred, as is the time in the victim's life: has there been a past outside the institution? Will there be a future? We know that her husband and two of her three children visited Maria a few times in Herzberge. The first two children, Franz and Christa, were allowed to accompany him, but as children, visits were only permitted from the age of 10; Hannah's father, Rainer, who was born in 1935, was never permitted to go with them. In the painting "*Die drei Besucher*" ("The three visitors", Figure 6), the visitors are depicted in the foreground and in front of them; in this wall of houses, Maria looks very small, almost disappearing into the cold light. The situation is gloomy. Between them is an

enormous, dark, empty space. This work is particularly interesting because it relates the outside world to the condition of the patients and refers to the question of the impossibility of "overcoming the external limit", symbolized by the building's walls, in the memory of the Nazi euthanasia crime. With very few exceptions, we have no direct testimonies of the victims.[23] Without an "internal voice" on the crime, we are blocked outside and forced to be distant from them and from their emotions.

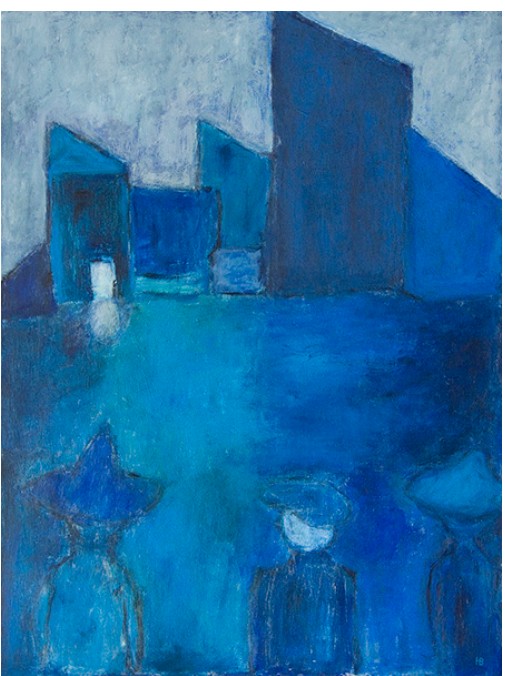

**Figure 6.** Hannah Bischof, "*Die drei Besucher*", acrylic on canvas, Berlin 2012, © Ben Bischof.

In the next work, "*Innenhof mit dunklem Bett*" ("Inner courtyard with dark bed"), Maria is transferred to the state institution in Neuruppin, Brandenburg, on 14 August 1941. This transfer is her death sentence: she will either starve to death, be given a lethal injection, or be taken by a grey bus to an Aktion T4 killing center. With her bed in the middle of the courtyard, surrounded by sharp, intimidating, coldly-colored houses, she becomes almost an object. She is alone, surrounded by an enormous, empty space: there are no other people, no other voices, and no one can understand her condition.

The cycle concludes with four works depicting the Neuruppin clinic. "*Die Klinik*" ("The Clinic"), "*Die Kapelle*" ("The Chapel"), and "*Die Särge*" ("The Coffins") can be considered a triptych; they have common features and the same colors. In the first one, Maria tries to escape and return to her family. She feels her time is running out and, in fact, in the next painting, we see her now dead and her body laid to rest in the Neuruppin chapel. The painted chapel appears transparent and glassy, behind it is the unknown. She was murdered on 7 August 1942, a week before her 37th birthday. At her husband's request, the corpse was brought to Berlin for burial. In the work that concludes the cycle of paintings, "*Kirchenschiff mit blauem Haus*" ("Church nave with blue house", Figure 7), the woman and her family are painted as houses: Josef with his three children, Franz, Christa, and Rainer, are represented as a church, while Maria is a small blue house, far from her family, alive only in their memory. Here too, the space that divides them is enormous, empty, and dark.

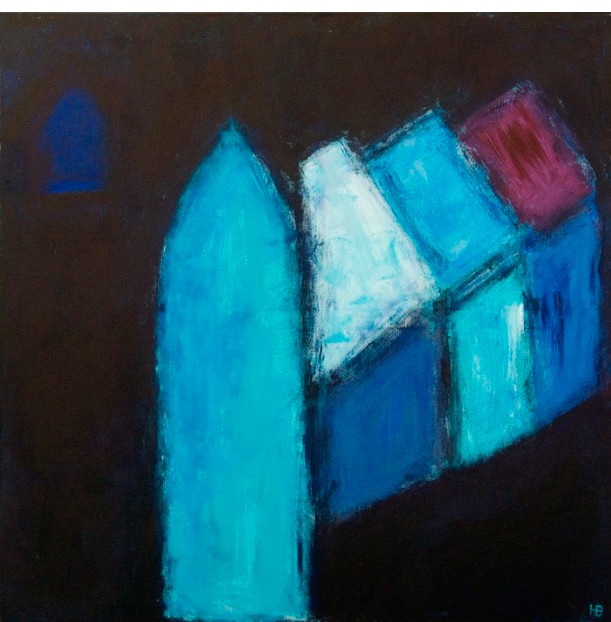

**Figure 7.** Hannah Bischof, "*Kirchenschiff mit blauem Haus*", acrylic on canvas, Berlin 2013, © Ben Bischof.

## 4. Conclusions

In a series of paintings produced over many years, Hannah, Maria's granddaughter, presents sixteen snapshots of her grandmother's life to narrate her story. This period of creative expression can be interpreted as a difficult attempt to process the family trauma internalized since childhood, by constructing something that could give it a material form: this is what I call a "memory object". The "Cycle for Maria" becomes a symbol with multiple meanings: it embodies the story of a victim of the Nazi Euthanasia Program but also the active role of her granddaughter, who enters the story by restoring, in a sense, the life the victim was denied and by opening up a new dimension for her, that of public memory.

"After the first exhibition, others followed, also in Papenburg, where Maria was born and lived, in Osnabrück, in the clinic where she was interned at the age of seventeen, and in the last exhibition this year in Neuruppin, where she was murdered. The paintings were exposed in the monastery church. It was very emotional for me, because I saw it as the closing of a circle. In a way it was as if Mary had returned there, no longer as a victim, but as a human being."[24]

Interestingly, the Cycle, as a "memory object" constitutes itself as an individuality, acting more as a subject than as an object: Hannah remarks how, thanks to her paintings, she became acquainted with many people, some of whom, after seeing her artwork, were prompted to start their own research and reconstruct a similar family story.

"I think it is important to show the human being, who is not a number among others in the total of the victims, but it was her! She was one of the victims. A woman who was killed for no reason, a woman who was also happy, who had children, who was a beautiful woman, and who then became ill, but could have been cured. I wanted to show that, I wanted people to know that. And every time there is an exhibition, some people come and talk to me, thank me, and tell me that what I do is important. One woman, for example, told me that she had a post-partum depression, like Maria, but that she was fine now. She was happy with her children and could talk openly about it. I think this pushed me more and more to want to talk about the illness, which is present in my family, about my grandmother's story, which is similar to the story of so many others. I think talking about it is important so that it can never happen again."[25]

Space becomes an element of major importance, but it is an empty space that seems impossible to fill and that surrounds the small and lost figure of Maria. The choice to portray the victim's story through the almost total absence of human elements is associated with the difficult formation of a shared memory of the Nazi Euthanasia Program. Among the reasons for this difficulty is the lack of "symbolic images" of the crime, necessary for collective memory in order to constitute itself and give rise to public discourse (Assmann 2011). In *De Oratore*, Cicero had already emphasized the close and necessary link between places and the "mental images" of what one wants to remember.[26] Centuries later, his initial reflection was taken up and articulated in an effective and more complex way by Maurice Halbwachs, with the concept of "images-souvenirs", and by Jan Assmann (Halbwachs 1950). In the specific case of the Nazi Euthanasia Program, the difficulty of finding images of the crime that collective memory can elect and use as "symbols" to self-define itself is combined with the almost total absence of a testimonial apparatus that can be considered as the internal voice.[27] In fact, only very few photographs and testimonies are available.[28]

The relatives of victims (specifically the third and fourth generation, who discovered the "family secret") decided to reconstruct the victim's history and share it through the creation of a "memory object". By accessing their own family archives, they were able to find images linked to the places of the victim's history to build a personal "collective family memory" based on the "places/images/identity group" structure, which is the basis for the formative process of cultural memory (Nora 1989, pp. 7–24). The ever-increasing number of literary and artistic works produced by the relatives of the victims, works of "post memory," as Marianne Hirsch would define them (Hirsch 2012), represent a radical change in the German cultural memory of this crime.[29] It is hoped that they will play the active role necessary for the creation of a real collective memory of Nazi euthanasia, which is still in the process of formation. By disseminating images from the family archives, which could become symbolic images, it will be possible to give a face and a name to at least some of the victims and to insert their stories into a network of physical places. Such places, being the backdrop of shared stories, will then fully become *lieux de memoire* (Nora 1984–1992).

**Funding:** This research was funded by German Academic Exchange Service, grant number 57552337; Auschwitz Foundation ASBL, Research Grant 2020-2021; Sapienza Università di Roma, Avvio alla Ricerca 2021; Sapienza Università di Roma, Avvio alla Ricerca 2022.

**Informed Consent Statement:** Informed consent was obtained from all subjects involved in the study.

**Data Availability Statement:** The data presented in this study are available on request from the corresponding author.

**Conflicts of Interest:** The author declares no conflict of interest. The funders had no role in the design of the study; in the collection, analyses, or interpretation of data; in the writing of the manuscript; or in the decision to publish the results.

## Notes

1    The first European state to enact such a law was Switzerland (Canton Vaud) in 1928, followed a year later by Denmark. Similar measures were enacted in Norway in 1934, in Sweden and Finland in 1935, in Estonia in 1936 and in Ireland in 1938. Vera Cruz, Cuba, Czechoslovakia, Yugoslavia, Lithuania, Latvia, Hungary and Turkey followed.

2    On the same day, the Concordat with the Catholic Church was also approved. To avoid complications with the Vatican, the law was not published until 25 July and took effect on 1 January 1934.

3    According to H. Friedlander in October 1939. (Friedlander 1995).

4    It was not published in the Official Gazette of the Ministry.

5    Ministry of the Interior Circular IV b 3088/39 1079 MIdR of 10 August 1939. Translation by the author.

6    As established by the Schwerin public Prosecutor's Office in 1949.

7    About the perception of the annihilation, it is impossible not to recommend the works of Primo Levi: (Levi 1956, 1958).

8    Interview with Hannah Bischof, Berlin, 9 November 2022, in Author's Private Archive (APA).

9    The interviews, audio and video recorded in Berlin by the author during 2019 and 2022, are conserved in the Author's Private Archive (APA). All interviews were conducted in English and German. The translation of the extracts is by the author.

10    I used the same interview-scheme as for narrative-biographical interviews. See in particular the works of Gabriele Rosenthal: (Rosenthal 1998, 1993, pp. 59–91). This interview technique works with an initial narrative question, which leaves the interviewee free to choose the main themes. This personal choice constitutes a valid first element for the subsequent analysis. In the second part, more specific questions are asked, but always with reference to the main narrative. In the last part, the interviewer asks questions concerning topics that did not emerge during the interview.

11    All extracts quoted in this paragraph are excerpts from the interview with Hannah and Gina, Berlin, 12 June 2019, in APA.

12    About the "special diet" see: (Burleigh 1994, p. 229 and following).

13    "*Die rote Mühle*" (The red mill)—Papenburg 1905, 2012, 30 × 40 cm; "*Die Abreise*" (The departure)—Osnabrück 1922/1923, 2012, 30 × 40 cm; "*Papenburg 1923–1927*"; 2015; 100 × 100 cm; "*Die Hochzeit*" (The wedding), 2015, 100 × 100 cm; "*Der Aufbruch*"—Umzug nach Hamburg 1927 (The start—Move to Hamburg), 2012, 100 × 100 cm; "*Ein neues Zuhause*"—Leben in Hamburg 1928 (A new home—Life in Hamburg), 2011, 100 × 120 cm; "*Gestörte Wahrnehmung*"—Staatskrankenanstalt Friedrichsberg/Hamburg 1928 (Disturbed Perception—State Hospital Friedrichsberg/Hamburg), 2015, 80 × 100 cm; "*Zwischenzeit*"—zwischen Hamburg und Berlin; Aufenthalte ohne Kliniken 1929–1938 (In-between—between Hamburg and Berlin; stays without clinics), 2015, 100 × 100 cm; "*Cardiazol*"—Sanatorium "Heidehaus"/Zepernick bei Berlin 1938, 2015, 100 × 100 cm; "*Die Konturen verschwimmen—Psychiatrie*"—Klinik Herzberge, Berlin-Lichtenberg 1938–1941 (The outlines blur—psychiatry), 2015, 100 × 100 cm; "*Die drei Besucher*"—1938–1941 (The three visitors), 2012, 120 × 160 cm; "*Innenhof mit dunklem Bett*"—Neuruppin 1941 (Inner courtyard with dark bed), 2014, 100 × 100 cm; "*Die Klinik*"—Neuruppin 1941/1942 (The clinic), 2015, 60 × 80 cm; "*Die Kapelle*"—Neuruppin 1942 (The Chapel), 2015, 40 × 60 cm; "*Die Särge*"—Neuruppin 1942 (The coffins), 2015, 40 × 60 cm; "*Kirchenschiff mit blauem Haus*"—Neuruppin/Berlin 1942 (Church nave with blue house), 2013, 100 × 100cm.

14    Translation by the author. The original, in German, can be found at this link: https://www.hannah-bischof.de/vita/uber-meine-malerei/, Consulted on 2 November 2022.

15    In this essay, only some of the paintings are shown. All works can be found on the artist's website: https://www.hannah-bischof.de/galerie/zyklus-fuer-maria/, consulted on 11 February 2023.

16    The German word *Abreise* indicates the general, theoretical concept of "departure".

17    The German word *Haustochter*, in use until the first half of the 20th century, indicates a young woman who lived with a foreign family for a certain period of time, to learn how to run the household.

18    See note 8.

19    See note 8.

20    See note 8.

21    The German word *Aufbruch* indicates the concrete, initial part of a departure process.

22    See note 8.

23    Worthy of note, however, are the testimony of (Manthey 1994). Never translated in English, it has been republished by Mabuse Verlag in 2021. See also (Kaufmann 1999).

24    See note 8.

25    See note 8.

26    Cicero, *De Oratore* II 86, 353–355, in (Cicero 1942, pp. 466–67): "*He inferred that persons desiring to train this faculty must select localities and form mental images of the facts they wish to remember and store those images in the localities, with the result that the arrangement of the localities will preserve the order of the facts, and the images of the facts will designate the facts themselves, and we shall employ the localities and images respectively as a wax writing tablet and the letters written on it.*"

27    In addition to the statements and documents of the perpetrators, such as the medical files compiled in the euthanasia program clinics, in some cases, after the war, relatives of the murdered and survivors approached the judiciary and reported on their experiences. Concerning Hadamar, for example, see (Schneider 2020).

28    About Elvira Manthey see also: (Robertson et al. 2019).

29    About the literary memorialization of the Nazi Euthanasia, see: (Knittel 2013, pp. 85–101).

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
