# Peer review of "Blurred Edges: Representation of Space in Transgenerational Memory of the Nazi Euthanasia Program"

_genealogy, doi:10.3390/genealogy7010019_

Round 1

Reviewer 1 Report

This article is important in addressing transgenerational remembrance of Aktion T4 and the legacy of the Nazi Euthanasia programme for one specific family. This is an under-researched area and it is good to see it being the subject of analysis.

However, as the article currently stands there are a number of areas of significant weakness that need to be substantially addressed and extensively revised by the author. I will take each section in turn. 

Introduction

A serviceable introduction to the history of T4 in relation to the life of Maria Fenski. However, a few key historical texts seem missing from the bibliography by researchers such as Henry Friedlander, Suzanne E. Evans and Michael Burleigh.

Given the visual dimension of the analysis, it was also surprising that there is no reference to existing efforts to visually represent or memorialise Aktion T4. E.g. at the T4 memorial Berlin; Gerhard Richter's painting Tante Marianne.

Transgenerational Memory in the family of the victim

This section makes extensive use of an interview with two members of Maria Fenski's family, but there is no discussion of the methodological or ethical approach taken to the interview, nor the specific rationale for interviewing two members of the family at the same time. 

Zyklus für Maria as a 'Memory Object'

At the moment the analysis in this section is disjointed, moving between the author's narration and commentary from the painter. The interpretative idea of the 'memory object' needs to be much more clearly articulated and used as the interpretative thread pulling together this section. In so doing this section would then present a clear, original interpretation of Hannah Bischof's art by the author to the reader. At present the cohesiveness of this interpretation is lacking. 

The recommended work on all of these sections would then allow for a more profound and enhanced conclusion.

Specific corrections

Page 3, Line 86: In this essay 

Page 7, Line 250: Should the age be 37 rather than 17? 

Reviewer 2 Report

Dear author

I really enjoyed reading and reviewing this important work, especially in the times we are living in now.  I made some comments in the script, comprising of reflections and thoughts that came to me while reading -  sometimes outrage at the contents - this is obviously my personal view and is not meant to be a provocation or a suggestion for the need to answer it. Just some thoughts.  

In other parts I have provided comments that are hopefully helpful for improving the text.

The most difficult part is the poor translation (many examples, here just one: 247) of the interview data in italics. This can be easily remedied with a pro translation tool or with ChatGPT. I have also noted some other areas where the translation of the main text can be improved.

I trust this is helpful. Thank you for this really important work.

Author Response

Thank you so much for your help! Please see the attachment.

Reviewer 3 Report

Thank you for the opportunity to review this fine article. I have one suggestion that the author can adopt or ignore. I am used to reading articles that begin with the historiography (literature review), so when the author placed it at the end of the article, it kind of threw me for a loop. But this certainly is not a valid reason to reject the article. This organizational scheme is personal preference. The historiography (literature review) is well done and includes the most relevant recent scholarship on the issue of second- and third-generation memory.

The story of Maria and her granddaughters' work to recover her story from available documentation is a significant one. This is the kind of work scholars must do as we continue to grapple with the Nazi regime's atrocities and how to remember the victims 80 years after the events. The author does a fine job providing a brief history of the euthanasia program and Maria's experiences within the program until her tragic death. The author skillfully weaves Maria's story with the story of Hannah and Gina, as they work to recover Maria's voice and help her tell her story. The author uses extensive quotes from the granddaughters. Generally, I would recommend reducing the number of quotes, but in this case I would argue that they are essential. This is an article about how descendants of a victim try to give voice to their relative so she can tell her story. To not use extensive quotes from the interviews would seem like an attempt to stifle the granddaughters' voices, and in turn, Maria's voice. Not only are the paintings accomplishing the work of giving voice to Maria, but so are the interviews with the granddaughters. They work in concert to help tell Maria's story.

I could see this as a documentary. With a little work, the article could be transformed into a documentary script. I really believe the author should explore this possibility. This would be a wonderful movie to use in a college classroom. The story is poignant and important.

I saw some very minor punctuation errors and other minor errors in the following lines.

1. Line 139 -- I assume it is meant to read "...to Berlin where two other children..."

2. Line 147 to 148 -- I assume it is meant to read "...September 1940, after the protest of her husband..."

3. Line 370 -- I believe the first part of the quotation marks around crazy are missing.

4. Line 450 -- buildings should be possessive. It is missing the apostrophe before the "s"

These are very minor things and definitely do not disqualify this article. I recommend the editors publish this article.

Author Response

I may sound repetitive because I have written this more than once in the attached document, but I would like to thank Reviewer 3 warmly. 

Round 2

Reviewer 1 Report

Improvements have been made to the literature review and the interpretative idea of the 'Memory Object'. However, from a methodological perspective, it is troubling that the explanation of the semi-structured interview methodology is thin (there isn't even a reference to an external source or authority on this research methodology). 

Author Response

Thank you for this additional suggestion. I have added, in a footnote, a reference to two works of Gabriele Rosenthal, who has extensively studied and used the interview technique that I use. On the same note, I have also summarized the phases of this type of interview. 

Thank you again for your important help in improving my article.